# Patients with Taste Disorders in a Hospital’s Dental Department and Strategies for Taste Disorders

**DOI:** 10.3390/biomedicines12092160

**Published:** 2024-09-23

**Authors:** Tatsuki Itagaki, Ken-ichiro Sakata, Taro Okura, Hirokazu Kobayashi, Sadasuke Hayata, Yoshimasa Kitagawa

**Affiliations:** Department of Oral Diagnosis and Medicine, Faculty of Dental Medicine, Graduate School of Dental Medicine, Hokkaido University, Kita-13 Nishi-7, Kita-ku, Sapporo 060-8586, Japan; titagaki@den.hokudai.ac.jp (T.I.); h.kobayashi@den.hokudai.ac.jp (H.K.); hayata@den.hokudai.ac.jp (S.H.); ykitagaw@den.hokudai.ac.jp (Y.K.)

**Keywords:** dysgeusia, phantogeusia, taste disorders, zinc acetate dihydrate, depression, aged

## Abstract

**Background/Objectives:** A retrospective study was conducted to clarify the clinical characteristics of taste disorder cases at the Department of Oral Medicine of Hokkaido University Hospital. The subjects were 322 taste disorder patients (86 male, 236 female, average age: 66 (13.1) years, mean duration of disorder: 15.2 (20.0) months) who were treated at our department from 2007 to 2018. **Methods:** Associations between symptoms and classification were examined. **Results:** When looking at the taste symptoms, 154 cases of quantitative taste disorder were observed as taste loss, abscission, and dissociative taste disorder, and 168 cases of qualitative taste disorder were observed as spontaneous abnormal taste, dysgeusia, and maltaste. There was no relationship between sex and quantitative/qualitative taste disorders at *V* = 0.08. When looking at the causes of taste disorders, the majority were psychogenic, idiopathic, and oral diseases. **Conclusions:** Approximately 20% of taste disorders are caused by oral diseases, so it should be noted that oral diseases such as oral candidiasis and xerostomia can cause taste disorders and that many of them can be improved with oral treatment.

## 1. Introduction

Taste is sensed by gustatory cells in taste buds located in the mucous membranes [1,2]. Taste receptor cells are mainly classified into type I, type II, and type III cell types, depending on their structure and gene expression [1,2]. There are two types of taste receptors, namely G protein-coupled taste receptors and ion channel taste receptors, which are sweet and bitter or salty and sour receptors, respectively [1,2]. Taste substances dissolve in saliva and, subsequently, are converted into gustatory signals that travel to the primary taste cortex [1,2]. The taste signal is transmitted to the hypothalamus and amygdaloid nucleus, which control emotions [1,2]. Furthermore, olfactory and gustatory information are integrated in the cerebral cortex, where they are recognized as food [1,2]. Therefore, taste disorders are thought to be associated with abnormalities in taste receptor mechanisms ranging from the periphery to the central nervous system. The number of patients visiting medical institutions with complaints of taste abnormalities is increasing year-by-year because of the stresses of modern society, an increase in the number of underlying diseases and commonly used drugs (polypharmacy), and the rapid progression towards a super-aging society [3]. Currently, zinc deficiency is considered to be one of the main causes of taste disorders in Japan, and reports from the otorhinolaryngology field indicate that the overall effectiveness of zinc treatments is approximately 70% [3,4,5,6,7]. However, the proportion of causes of taste disorders differs from that reported by otorhinolaryngology, and our department has experienced many cases in which zinc supplementation therapy was not effective [8,9]. Since the treatments of diseases differ depending on the clinical department that treats them, it is necessary to clarify the characteristics of the patients being treated. Many of the existing case reports of taste disorders are related to COVID-19, cancer, and drugs [10,11,12]. Oral diseases that cause abnormalities in taste include oral candidiasis, xerostomia, and stomatitis. Oral candidiasis is an opportunistic infection characterized by the abnormal growth of *Candida* and is characteristically common in elderly people [13]. *Candida* colonizes the mucous membranes on the tongue’s surface and inhibits the diffusion of tastants [14]. In addition, *Candida* invasion of mucous membranes causes damage to the peripheral nervous system. It is believed that taste abnormalities develop through these mechanisms. There are also other causes; for example, some patients with severe periodontitis complain of a bitter taste from the inflammatory exudate. Another example is xerostomia, which is broadly classified into peripheral and central and is often caused by insufficient water intake or salivary gland disease. However, it may also be caused by central diabetes insipidus. Additionally, stomatitis is frequently caused by malnutrition, inflammatory diseases, such as Crohn’s disease, and gastrointestinal diseases. While not in itself a taste disorder, the pain of stomatitis may make it difficult to recognize the true taste of food. Moreover, the malabsorption of nutrients due to gastrointestinal disease is another cause of taste disorders. Due to these myriad causes, patient interviews and examinations by multiple medical departments may be necessary. Although there are case reports of taste disorders in the field of otorhinolaryngology [7], there are almost no reports from dentistry. Most of the previous research elaborated on one or a few aspects of interest [4,10,11,12]. Therefore, the purpose of this study was to examine the characteristics of taste disorders in dentistry from a retrospective perspective, mainly regarding the subjective symptoms of taste disorders and the causes of taste disorders. Understanding the etiology will help us to develop treatment strategies.

## 2. Materials and Methods

### 2.1. Taste Test

The true purpose of taste tests is to evaluate whether taste buds function as receptors, whether the nerves transmit taste, and whether the brain can perceive taste.

For the assessment of gustatory sensitivity, the filter paper disc (FPD) method, which is covered by health insurance, was used in our hospital. FPD, which uses a 5 mm diameter piece of filter paper soaked in a taste solution, can evaluate the cognitive threshold of four taste regions of the chorda tympani, glossopharyngeal, and greater petrosal nerves, respectively [6]. We stimulated the test area for 3 s using taste solutions at different ascending concentrations and measured the cognitive threshold [15,16]. For a long time, Taste Disc^®^ was commonly used as a FPD tool, but it has been discontinued and is no longer available on the market as of February 2022. The test was therefore conducted using reagents made at our own facility. FPD uses 5 levels for 4 taste stimuli (Table 1). A high score on FPD indicates a lowered perceptual sensitivity. Elderly people generally have an increased taste perception threshold; therefore, the reference range changes depending on age.

In the whole mouth method, four types of taste solutions were dropped onto the dorsum of the tongue at the following concentrations (Table 2), and the cognitive thresholds were tested by tasting the samples for 2 to 3 s. Taste perception threshold testing followed the criteria in the table below; when No. 5 could not be recognized, taste loss was diagnosed. The reference ranges for these qualitative tests vary depending on the facility, and the normal values for cognitive thresholds vary greatly from person to person, so they alone are not sufficient for evaluating taste disorders. Therefore, a comprehensive diagnosis must be made.

### 2.2. Patients

This retrospective study included 322 patients with a history of taste complaints (mean duration: 15.2 (20.0) months) who were diagnosed with taste disorders at Hokkaido University Hospital between 2007 and 2018. In Japan, oral medicine has been established as a subspecialty of oral surgery. At our facility, the patients were diagnosed based on the criteria listed in Figure 1. All of them were diagnosed by an oral surgeon specialist with over 10 years of experience who was accredited by the Japanese Society of Oral and Maxillofacial Surgeons (S.K.-i.). There were 86 male patients and 236 female patients, with an average age of 66.3 (13.1) years. Since most patients who visit our facility are female, there was no imbalance in taste abnormalities based on sex.

### 2.3. Zung Self-Rating Depression Scale (SDS)

The Zung self-rating depression scale (SDS) is an instrument that screens for depression. The SDS is available in many languages. The reliability and validity of the SDS has been demonstrated in Japanese.

### 2.4. Ethical Aspects

The study data and informed consent were obtained in accordance with the Declaration of Helsinki, and the study protocol was approved by the Ethics Review Board of Hokkaido University Hospital (Approval No. 018-0381; approved in June 2019).

### 2.5. Statistical Analysis

The relationship between sex and taste disorder classification was examined using Cramer’s coefficient of association (*V*). A relationship was defined when *V* was ≥ 0.3. A relative evaluation was performed by calculating percentages from the observed frequencies for each category. The data were analyzed using Excel (Microsoft^®^ Excel^®^ for Microsoft 365MSO (version 2306, build 16.0.16529.20164, 64 bit)) and R version 4.3.1 (16 June 2023) (copyright © 2023, The R Foundation for Statistical Computing).

## 3. Results

As shown in Table 3, there were no imbalances in age and serum zinc levels between the two groups (Appendix A). Moreover, there was no relationship between sex and taste disorder classification (*V* = 0.082, 95% confidence interval: 0.005–0.19) (Appendix A). The differences between quantitative and qualitative taste abnormalities, classified by their causes, are shown in Figure 2. Qualitative taste abnormalities were 1.5 times more likely to be psychogenic and zinc deficiency-related than quantitative taste abnormalities. Prior to the coronavirus outbreak, most cases were caused by psychogenic causes, oral diseases, and idiopathic diseases. The most common oral disease was glossitis, caused by oral candidiasis.

## 4. Discussion

These results showed that there was no relationship between taste abnormalities, age, sex, and serum zinc levels (Appendix A). To our knowledge, there were no reports showing a relationship between sex and the cause of taste abnormalities. Regarding serum zinc levels, no relationship was observed, consistent with the trend in the two-category data from Nin et al. [7,15]. It is possible that zinc is only associated with certain taste abnormalities. Compared to the results compiled by Nin et al., common cold- and systemic disease-related taste disorders were twice as rare in our results, and drug-induced cases were five times less common [7,15]. On the other hand, cases due to oral diseases were 3.6 times more common [7,15]. These differences may be due to patients and referring physicians using appropriate judgment to select appropriate departments for hospitalization and treatment.

A comprehensive diagnosis should be made by utilizing taste recognition threshold tests, blood tests, and visual inspections. In our study, taste disorders were classified by symptoms and by cause.

Taste disturbances can be classified into four main categories: hypogeusia (a decreased sensitivity to taste modalities), dysgeusia (taste confusion), phantogeusia (phantom taste), and ageusia (loss of taste) [15,16]. The total loss of taste is rare. Hypogeusia, dysgeusia, and ageusia can be detected using the FPD method. Quantitative taste abnormalities constitute either a difficulty in detecting or an inability to detect some or all tastes. These can also be quantitatively evaluated using the FPD method. Qualitative taste abnormalities are taste disorders that have other symptoms. Since qualitative taste abnormalities are difficult to objectively assess, they are often assessed by asking patients about their subjective symptoms. As an alternative, cause-related classification allows for the selection of different cause-based therapies as treatments (Figure 1). There are also many causes that depend on the location of the disorder for classification; taste disorders generally occur at conductive, receptor, and neurological sites. However, this classification is incomplete because sometimes the cause is unknown.

In Japan, although oral zinc therapy was considered to be low-level evidence in a previous clinical review, polaprezinc has been used to treat taste disorders since 2011, with approval granted for its off-label use [4,7,8,15,16]. Polaprezinc is zinc acetate dihydrate, which is also available in Europe. Other treatment options are not covered by health insurance. Our previous study showed that ethyl loflazepate monotherapy could also be a treatment option for patients with idiopathic and psychogenic taste disorders [9].

In this study, psychogenic taste disorders accounted for 30–40% of cases, zinc deficiency-related taste disorders accounted for approximately 10%, and oral diseases accounted for 20% of cases. Based on the case series and descriptive statistics in this study, approximately 40–50% of the patients could be treated using polaprezinc or ethyl loflazepate. If oral disease can be ruled out as a cause based on the breakdown of causes of taste abnormalities experienced by patients visiting a dental clinic, the best course of action would be to explore the cause while taking zinc supplements. However, according to the cause-based classification, more than half of the cases required treatments other than zinc supplementation. The Zung self-rating depression scale (SDS) could be useful as a diagnostic aid to help with cause-based classification. It is useful for screening for psychogenic or depressive disorders; therefore, it can be calculated that using SDS as a diagnostic aid for psychogenic taste disorders would be effective for approximately one in four people (113/322 × 0.75 = 0.263).

In Japan, it is believed that zinc supplementary therapy is an effective treatment for taste disorders [3,4,12,15,16]. Based on the results of the study and the fact that no relationship was found between the type of taste disorder and serum zinc level, serum zinc levels constitute only reference-level information. Ingredients other than zinc in polaprezinc may be involved in the treatment of taste disorders [8]. Ito et al. considered carnosine to be one of the active ingredients with a treatment effect [8]. Considering previous studies as well as our study, it is possible that zinc and polaprezinc do not affect all taste disorders. These results may be due to changes over time rather than the drug. Since there are no drugs that are guaranteed to be effective against taste disorders, it is desirable to identify the causative disease and treat it. Therefore, based on the clinical results herein, further development of this basic research is desired.

A research limitation was the research design, which comprised a cross-sectional study. Without a cohort study, the effects of any interventions cannot be investigated.

The criteria for determining treatment results were ambiguous. Some patients complained of taste abnormalities even if their taste perception thresholds were normal. However, we sometimes need to deal with the discrepancy between a patient’s symptoms and objective results. Therefore, a better evaluation is desirable. For example, a Likert scale is a symmetrical evaluation system with five, seven, or nine points, comprising answers such as improved, slightly improved, unchanged, slightly worsened, and worsened. Likert scales would therefore be useful as a unified evaluation criterion for taste abnormalities and disorders.

## 5. Conclusions

It is important to identify the causes of taste disorders, as their treatment varies depending on the cause. Patients with taste disorders who visit dentists have a high proportion of psychogenic and oral diseases. Approximately half of these patients may require zinc supplementation therapy. Further research should establish standardized treatment evaluation criteria. In addition to testing the taste perception threshold, a Likert scale would be a suitable evaluation criterion. We also should provide appropriate treatments.

## Figures and Tables

**Figure 1 biomedicines-12-02160-f001:**
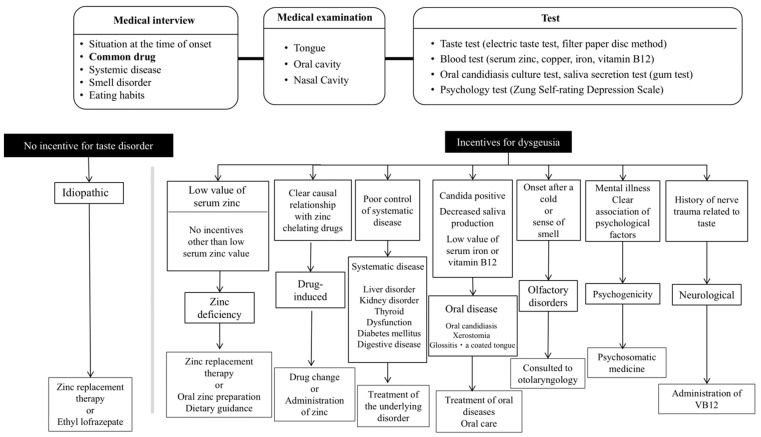
Diagnosis and treatment of taste disorders. At our facility, diagnosis and treatment are performed according to this flow diagram.

**Figure 2 biomedicines-12-02160-f002:**
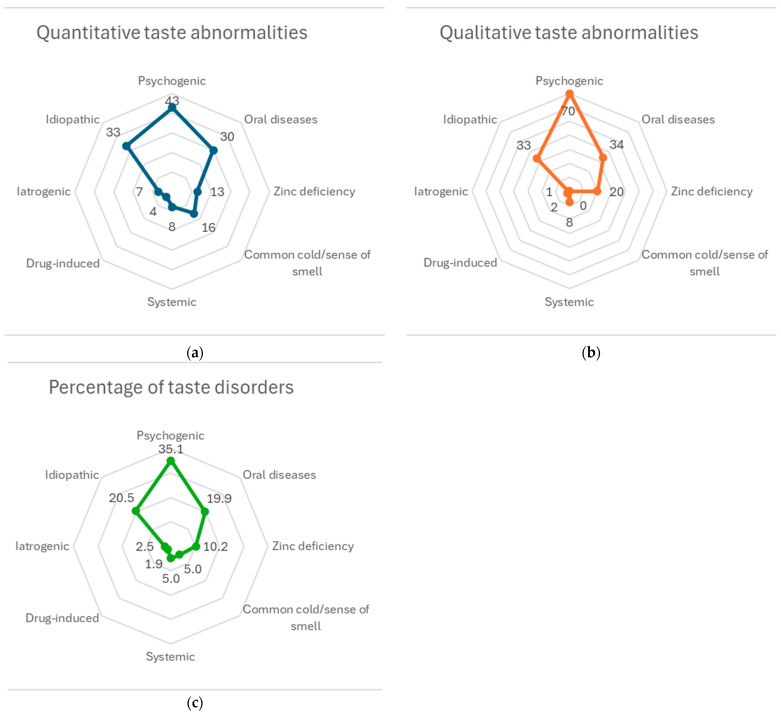
Aggregated results by taste disorder cause classification. The number of cases of quantitative and qualitative taste abnormalities and the combined ratio of both are shown in (**a**), (**b**), and (**c**), respectively.

**Table 1 biomedicines-12-02160-t001:** Filter paper disk (FPD) test.

Taste Disc^®^	No. 1	No. 2	No. 3	No. 4	No. 5	Molecular Weight
Sucrose: S	0.3%	2.5%	10%	20%	80%	342.2965
Sodium Chloride (NaCl): N	0.3%	1.25%	5%	10%	20%	58.44
Tartaric Acid: T	0.02%	0.2%	2%	4%	8%	150.0868
Quinine Hydrochloride Dihydrate: Q	0.001%	0.02%	0.1%	0.5%	4%	396.9083

Sucrose: S; sodium chloride (NaCl): N; tartaric acid: T; quinine hydrochloride dihydrate: Q.

**Table 2 biomedicines-12-02160-t002:** Whole mouth method.

	No. 1	No. 1.5	No. 2	No. 2.5	No. 3	No. 4	No. 5
S	2.9 μmol	5.8 μmol	8.8 μmol	17.5 μmol	29.2 μmol	58.4 μmol	2337 μmol (80%)
N	3.4 μmol	5.1 μmol	6.8 μmol	10.3 μmol	17.1 μmol	85.6 μmol	3422.3 μmol (20%)
T	0.3 μmol	0.7 μmol	2.0 μmol	6.7 μmol	10 μmol	20 μmol	533 μmol (8%)
Q	2.5 pmol	13.0 pmol	50.0 pmol	130 pmol	250 pmol	1.3 μmol	101 μmol (4%)

Sucrose: S; sodium chloride (NaCl): N; tartaric acid: T; quinine hydrochloride dihydrate: Q.

**Table 3 biomedicines-12-02160-t003:** Patient demographic data.

Taste Disorder Classification	Quantitative Taste Abnormalities	Qualitative Taste Abnormalities	Standardized Difference
Male	47	39	0.16
Female	107	129	
Age (mean (SD)) year	67.3 (14.7)	65.5 (11.4)	0.13
Age (median (range)) year	67 (24–90)	72 (18–94)	Not applicable
Zn (mean (SD)) μg/dL	73.1 (16.3)	73.4 (15.8)	0.02

Standard deviation: SD.

## Data Availability

The data presented in this study are openly available in this paper.

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
