# Peer review of "Patients with Taste Disorders in a Hospital’s Dental Department and Strategies for Taste Disorders"

_biomedicines, 2024, doi:10.3390/biomedicines12092160_

Round 1
Reviewer 1 Report
Comments and Suggestions for Authors
Dear Authors,
major revisions are required.

Comments on the Quality of English LanguageDear Authors,
extensive editing of English language is required.
Reviewer 2 Report
Comments and Suggestions for Authors
Dear authors, thank you for the submitted article. The article is interesting but I don't know how practical it is. Despite the title, it concerns the frequency of taste disorders. According to the authors, it is mainly caused by somatic disorders, and zinc deficiency. It was pointed out that zinc supplementation is a common practice, including the use of Polaprezinc preparations, which are not approved in Europe. It is a pity that the authors did not focus more on oral diseases, because it would be more useful for practicing physicians.
As representatives of dental medicine, you have paid relatively marginal attention to oral diseases. You have classified diseases such as Candidosis, xerostomia, and glossitis into one group. Why? In the conclusion (which sounds different in the text and in the abstract) you wrote that 20% of cases are related to oral diseases. It is very interesting how taste disorders depend on the stage of mycosis (according to Newton). Also noteworthy is the fact that the article's title does not fully reflect its content - which is also reflected in the conclusion.
Round 2
Reviewer 1 Report
Comments and Suggestions for Authors
Dear Authrs,
thank you for your detailed answers.
Yours sincerely,
Reviewer
Comments on the Quality of English LanguageDear Authors,
a moderate editing of the Englisg language is required.
Yours sincerely,
Reviewer
Author Response
Dear Reviewer 1,
Thank you very much for reviewing our manuscript and offering valuable advice.
Our maniscript was difficult to read, but we have used the English Editing MDPI service and revised it to make it easier to read because we want our manuscript to be read by many readers.
Again, thank you for reading the ambiguous and difficult-to-read text and pointing out the appropriate points. Thanks to you, our manuscript has improved. We have worked hard to incorporate your feedback and hope that these revisions persuade you to accept our submission.
Thank you.
Sincerely,
Reviewer 2 Report
Comments and Suggestions for Authors
For me is ok
Author Response
Dear Reviewer 2,
Thank you very much for reviewing our manuscript and offering valuable advice.
Our maniscript was difficult to read, but we have used the English Editing MDPI service and revised it to make it easier to read because we want our manuscript to be read by many readers.
Again, thank you for reading the ambiguous and difficult-to-read text and pointing out the appropriate points. Thanks to you, our manuscript has improved. We have worked hard to incorporate your feedback and hope that these revisions persuade you to accept our submission.
Thank you.
Sincerely,
Round 3
Reviewer 1 Report
Comments and Suggestions for Authors
Dear Authors,
thank you for the revised version of the manuscript.
Yours sincerely,
Comments on the Quality of English Language
Dear Authors,
thank you for making your manuscript readable.
Yours sincerely,